# Pro-Health Potential of Fruit Vinegars and Oxymels in Various Experimental Models

**DOI:** 10.3390/ijms26010007

**Published:** 2024-12-24

**Authors:** Beata Olas

**Affiliations:** Department of General Biochemistry, Faculty of Biology and Environmental Protection, University of Lodz, Pomorska 141/3, 90-236 Lodz, Poland; beata.olas@biol.uni.lodz.pl; Tel./Fax: +48-42-6354485

**Keywords:** biological activity, fruit vinegar, oxymel, pro-health potential, submerged fermentation

## Abstract

Fruits are excellent sources of substrate for various fermented products, including fruit vinegars, which are typically produced by submerged fermentation. Some evidence suggests that fruit vinegar consumption can alleviate certain disorders, including hyperlipidemia, inflammation, and hyperglycemia. Fruit vinegars also have bacteriostatic and antihypertensive actions. Recent studies also suggest that apple vinegar may offer benefits in treating insulin resistance, osteoporosis, and certain neurological diseases such as Alzheimer’s disease; it may also support weight loss. Recent studies in animal and human models have considerably broadened our understanding of the biological properties of not only fruit vinegars but also oxymels, i.e., mixtures of vinegar and honey or sugar. This paper reviews the current state of knowledge regarding vinegars and oxymels, with a special emphasis on their chemical composition and the mechanisms behind their biological activity and pro-health potential. The multidirectional effects of fruit vinegars and oxymels result from the synergy of different chemical compounds, including organic acids (mainly acetic acid), phenolic compounds, vitamins, minerals, and fermentation products. However, more studies are needed to understand the interactions between all the different components, not only the phenolic compounds and organic acids. In addition, more research is needed on their mechanisms of action. Although no serious side effects have been noted to date, further studies with large sample sizes are needed to understand the possible side effects of long-term fruit vinegar and oxymel use.

## 1. Introduction

Health-promoting foods have recently enjoyed increasing interest from consumers. Fruits, for example, are valuable sources of vitamins, minerals, and fiber and contain a range of phytochemicals, such as phenolic compounds, known to benefit health by exerting anti-inflammatory, hypoglycemic, antioxidant, and anti-platelet activity. A range of apples are popular choices for producing a range of alcoholic beverages and vinegars. Most fruit vinegar production is based on aromatic and juicy apple species, such as McIntosch, Cox Orange, Boskop, and Alkmene [1,2,3,4,5]. However, fruit vinegars can also be made from many other fruits, including grapes, blackberries, strawberries, oranges, pineapples, and bananas, as well as fruit juices. They can be prepared by a two-stage process based on alcoholic and acetic acid fermentation.

Food fermentation is one of the oldest forms of biotechnology and was initially used as a means of preservation. Currently, there are about 5000 varieties of fermented food products [3,6,7], and they can have unique characteristics, tastes, and flavors depending on the ingredients, the fermentation process, and the participating microorganisms.

A key determinant of vinegar quality is its proportions of specific compounds, particularly phenolic compounds. However, the most important criterion for assessing the quality of vinegar, along with the aroma, is its acetic acid content. In addition, an important part is played by the sugar content in the fruit as this determines the amount of alcohol produced and thus the acetic acid content. The acetic acid content should not be less than 5–7% in fruit vinegars, while the maximum ethanol content should not exceed 1%. There are several ways to obtain vinegars, for example, the Orleans method (one of the oldest methods, i.e., the surface method), the generator drip method, or the depth method [2,8,9,10,11,12,13,14].

According to historical records, fruit vinegars were first prepared and utilized by the ancient Egyptians, Babylonians, and Sumerians. In addition to their use as taste enhancers, they have also been employed as medicines: the ancient Egyptians used apple vinegar to treat mushroom poisoning and a lack of appetite, while Hippocrates recommended vinegar as a disinfectant for wounds and apple vinegar for treating colds, coughs, and digestive ailments. They began to be used medicinally by Europeans in the 17th century; however, currently, in Africa, America, and Europe, fruit vinegars are mainly added to food for their flavor [15]. They have also been included in syrups and antiseptics as an antimicrobial agent [8,12].

Today, fruit vinegars are popular natural products with multiple uses and are often used as dressings, particularly in salads. Apple vinegar is becoming particularly popular, mainly because of its organoleptic profile and its phenolic compound profile, which is believed to confer various biological properties [2,4,5].

Chemically speaking, fruit vinegars are aqueous solutions of acetic acid, with a mixture of inter alia organic acids, mineral salts, dyes, esters, aldehydes, and ketones [2,13]. They are manufactured using various production techniques, the choice of which determines their taste, color, consistency, chemical composition, smell, or biological properties. Their production process is carried out in two stages. In the first stage, anaerobic digestion takes place, in which sugars are converted into ethanol with the participation of yeast. In the next step, ethanol is converted to acetic acid with the participation of acetic acid bacteria (AAB) from the *Acetobacteriaceae* family, including *Acetobacter*, *Komagataeibacter*, *Gluconacetobacter*, and *Gluconobacter* [2,13,16]. More details about the microorganisms used for fruit fermentation are described by Yuan et al. [7].

The consumption of fruit vinegars has been positively associated with good health. Fruit vinegar consumption is associated with inter alia better control of diabetes, blood pressure, and obesity, associated with better lipid metabolism [2,8,17,18,19,20,21,22,23,24,25,26]. In addition to fruit vinegars, recent studies have considerably broadened our understanding of the biological properties of oxymels, i.e., mixtures of vinegar and honey or sugar. They can be used in a simple form or enriched with various medicinal plant roots, seeds, leaves, fruits, and vegetables extracts. One widely used oxymel is squill oxymel, which has gained recognition in the West and continues to be used today [27].

The term *oxymel* is derived from the Greek words *oxymeli*, meaning *acid and honey*. It has been used for centuries as a therapeutic drink with a unique flavor and is still a commercially produced traditional drink in various countries. The use of oxymel dates back to the time of Hippocrates [8,27]. In ancient Persia, oxymel was known as *serkangabin*, derived from the combination of *serkeh* (vinegar) and *angabin* (honey). Persian medicine books describe various types of oxymel and their preparation methods, side effects, benefits, and indications; they include over 1200 types of oxymels, including one containing apple vinegar and honey, often named *powerdrink* [27].

Recently, few papers have described the chemical characteristics and biological properties of various vinegars [12,14,26]. However, Ousaaid et al. [2] only demonstrated the chemical composition of fruit vinegars and their therapeutic application. In addition, only one systematic review on oxymel has been published [27]. The present work reviews the up-to-date literature concerning the pro-health potential of fruit vinegars, especially filtered and pasteurized clear vinegars, and oxymels; it also examines their chemical composition and the mechanisms behind their biological properties. Of course, artisanal vinegars, i.e., live vinegars, are also available, but no comprehensive literary data on their health-promoting potential currently exist.

## 2. Methodology for Literature Search

A literature search was performed of PubMed, Science Direct, Scopus, Springer, Web of Knowledge, Web of Science, and Google Scholar, using various combinations of the keywords “fruit vinegar”, “oxymel”, “biological activity”, and “pro-health potential”. No time criteria were applied to the search, but recent papers were evaluated first. The review included book chapters, review papers, and research papers. The abstracts of any identified articles were initially analyzed to confirm whether they met the inclusion criteria. Any relevant identified articles were summarized. After obtaining the full texts of the included studies, the reference sections were also manually examined to identify any additional new articles.

## 3. Phytochemical Characteristics of Fruit Vinegar and Oxymel

### 3.1. Fruit Vinegar

Fruit vinegars contain a cocktail of chemical compounds. The main components are organic acids, particularly acetic acid, which makes up about 30–50% of the total organic acid content. These are accompanied by melanoidins formed between sugars and nitrogen compounds such as amino acids, peptides and proteins, and tetramethylpeyrazine, also known as ligustrazine. In addition, vinegars contain a range of phenolic compounds, minerals (e.g., sodium, potassium, calcium, zinc, iron, copper, phosphorus, and magnesium), vitamins (A, C, and others), amino acids, monosaccharides (e.g., glucose, fructose, xylose, mannitose, and arabinose), disaccharides (such as sucrose, maltose, and mycose), and pectin (Figure 1).

They contain free and nonprotein amino acids, mainly derived from raw materials and proteins obtained by microbial decomposition, and their total concentration increases with aging time. Glucose and fructose may be present, and they provide a sweet taste to fruit vinegars. A comparison of the chemical compositions of different fruit vinegars is shown in Table 1. For example, apple vinegar has 24–226 mg/L Br and Ca, 19.4 mg/L Fe, and 7–195 mg/L Mg [2,13,15].

#### Phenolic Compounds and Organic Acids

In addition to phenolic compounds, organic acids are also believed to play an important part in the beneficial effects of fruit vinegars [28].

The phenolic compound composition of a fruit vinegar is influenced by its raw material [29,30,31,32,33,34]. For example, chlorogenic acid predominates in apple vinegar [29], and “Rauch”-made vinegar has the highest total phenolic compound content (281 mg gallic acid equivalent (GAE)/L) among commercial apple vinegars (33–57 mg GAE/L) [35]. The content of phenolic compounds obtained from dark fruits ranges from 367.2 mg GAE/L in raspberry vinegar to 1443.6 mg GAE/L in cherry and elderberry vinegar [36]. In grape vinegars, the content ranges from 71.01 to 2228.79 mg GAE/L [11].

A recent study by Abdali et al. [37] found that the chemical composition of apple vinegar, including acetic acid and phenolic compound content, depended on the choice of apple cultivar. A study of fruit vinegars available on the Polish Food Market (apple, rhubarb, lemon, and pear vinegar) by Melkis and Jakubczyk [38] found lemon vinegar to have the highest vitamin C content (15.95 mg/100 mL) but apple vinegar to have the highest flavonoid content (70.22 mg RE/L). Ozdemir et al. [39] note that sea buckthorn fruit vinegar is also a good source of phenolic compounds such as gallic acid (763.9 mg GAE/L) and that the major phenolic compound in *Rosa canina* L. vinegar is catechin (5.66 mg/L). It also appears that the phenolic compound content of a vinegar may be related to its antioxidant activity [36].

Suksamran et al. [40] report that mangosteen rind vinegar, mangosteen flesh vinegar, and mangosteen rind plus flesh vinegar have higher levels of phenolic compounds and greater antioxidant activity than apple vinegar. Also, a recent study by Uram-Dudek et al. [36] found dark fruit vinegars to have higher antioxidant activity, i.e., 21.3 to 77.5% tested by the 2,2-diphenyl-1-picrlhydrazyl (DPPH) method, compared with other vinegars [36].

Budak et al. [41,42] report the total phenolic content to be 1483.66 mg GAE/L in grape juice and 2690 and 2461 mg/L in vinegar obtained by traditional and industrial methods, respectively; they also indicate that fruit vinegars have higher antioxidant activity than wines and fruit juices. In addition, their bioaccessibility may be decreased or increased by fermentation. For example, the gallic acid and *p*-hydroxybenzoic acid in commercial “Turkish” apple vinegar were found to be less bioaccessible than those in apple fruits [43]. However, no other studies have examined the bioavailability of the other phenolic compounds present in fruit vinegars.

Nevertheless, a range of antioxidant compounds are present in fruit vinegars, and these can derive from the fruits or arise during the fermentation process [43,44]. The antioxidant 1,4-lactone of D-saccharic acid is also formed during fermentation [45], demonstrating that the method of production can also influence the final antioxidant potential [41,42,46]. Also, Antoniewicz et al. [47] monitored the antioxidant activity and phenolic compounds content of grape vinegars during the two-month fermentation process and the subsequent six-month storage under various conditions. It was found that storage conditions and time both affected the phenolic compound content and thus the antioxidant potential of the vinegar; the content was also influenced by the grape variety and the preparation method.

While the characteristic aroma of vinegar is attributed to the presence of acetic acid, it may also be influenced by a variety of chemical compounds formed during acetic acid fermentation [39]. For example, Budak et al. [41,42] report that peach vinegar has high concentrations of acids (68%), alcohols (10%), and esters (8%), which contribute to its odor. More details about chemical characteristic of fruit vinegars are given in reviews by Xia et al. [25] and Ousaaid et al. [2].

**Table 1 ijms-26-00007-t001:** Comparison of selected chemical ingredients in the different fruit vinegars.

Selected Chemical Ingredients	Fruit Vinegar										
	Pomagranate	Blackberry	Blueberry	Mulberry	Cherry	Apple	Plum	Kiwi	Grape	Persimmon	Pineapple
Amino acids	Aspartic acid, glutamic acid, serine, alanine, glycine, arginine, threonine, lysine, valine, isoleucine, methionine, leucine, phenylalanine, and GABA [48]	Aspartic acid, glutamic acid, serine, alanine, glycine, arginine, threonine, lysine, valine, isoleucine, methionine, leucine, phenylalanine, and GABA [48]	Aspartic acid, glutamic acid, serine, alanine, glycine, arginine, threonine, lysine, valine, isoleucine, methionine, leucine, phenylalanine, and GABA [48]	Aspartic acid, glutamic acid, serine, alanine, glycine, arginine, threonine, lysine, valine, isoleucine, methionine, leucine, phenylalanine, and GABA [48]	Aspartic acid, glutamic acid, serine, alanine, glycine, arginine, threonine, lysine, valine, isoleucine, methionine, leucine, phenylalanine, and GABA [48]	-	-	-	-	-	-
Sugars	Fructose and glucose [48]	Fructose and glucose [48]	Fructose and glucose [48]	Fructose and glucose [48]	Fructose and glucose [48]	-	-	-	-	-	-
Organic acid	Barbituric acid, shikimic acid, citric acid, succinic acid, and acetic acid [4]	Maleic acid, barbituric acid, shikimic acid, adipic acid, citric acid, succinic acid, lactic acid, acetic acid, and propionic acid [4]	Malonic acid, barbituric acid, quinic acid, shikimic acid, citric acid, tartaric acid, malic acid, succinic acid, lactic acid, and acetic acid [4]	-	Isobutyric acid, isovaleric acid, hexanoic acid, octanoic acid, nonanoic acid, decanoic acid, dodacanoic acid, tetradecanoic acid, methyl acetate, ethyl acetate, ethyl propanoate, isobutyl acetate, isoamyl acetate, ethyl caproate, ethyl caprylate, ethyl decanoate, benzyl acetate, phenethyl acetate, ethanol, isobutyl alcohol, hexanol, nonanol, and benzyl alcohol [4,49]	Acetic acid, lactic acid, quinic acid, malonic acid, barbituric acid, oxalic acid-dihydrate, shikimic acid, adipic acid, oxalic acid, tartaric acid, propanedioic acid, malic acid, succinic acid, propionic acid, isobutyric acid, butryric acid, isovaleric acid, and citric acid [4,29,30,50]	Acetic acid, tartaric acid, and lactic acid [51]	Acetic acid, lactic acid, quinic acid, tartaric acid, propanedioic acid, malic acid, succinic acid, and citric acid [29]	Malonic acid, barbituric acid, shikimic acid, adipic acid, citric acid, tartaric acid, succinic acid, lactic acid, acetic acid, fumaric acid, and propionic acid [4]	-	Methyl ester, ethyl acetate, isobutyl acetate, iobutanol, acetoin, benzaldehyde, propanoic acid, butanoic acid, isobutyric acid, methylbutanoic acid, naphthalene, and phenylrthyl alcohol [52]
Phenolic compounds	Gallic acid, galloylglucoside, protocatechuic acid, punicalagin, catechin, vanillic acid, syringic acid, ethyl acid, chlorogenic acid, caffeic acid, *p*-coumaric acid, ferulic acid, ferulic acid hexoside, tyrosol, and *trans-p*-coumaric derivate [53,54]	-	-	-	Gallic acid, chlorogenic acid, *p*-coumaric acid, caffeic acid, ferulic acid, catechin, protocatechuic acid, caftaric acid, furoic acid, protocatechualdehyde, tyrosol, catequin, vanillic acid, syringic acid, vanillin, syringaldehyde, coniferyl aldehyde, sinapaldehyde, and epicatechin [49,55]	Chlorogenic acid, 4-coumaroylqunic acid, isomer of chlorogenic acid, isomer of 4-coumaroylqunic acid, *p*-hydroxybenzoic acid, protocatechuic acid, gallic acid, vanillic acid, caffeic acid, *p*-coumaric acid, trans-ferulic acid, catechin, syringic acid, epicatechin, gallate, procyanidin B2, luteolin-3-*O*-rutinose; isorhamnetin-3-*O*-rutionse, isorhamnetin-3-*O*-glucoside, kaempferol-3-*O*-glucoside, quercetin-3-orhamnoside, quercetin, rutin, luteolin, apigenin, phloretin, phloridzin, and phloridzin [29,30,53,56,57,58]	Ellagic acid, caffeoylquainic acid derivatives, *p*-coumaric acid derivatives, cyjandidn 3-*o*-galactoside, cyjanidin 3-orobinobioside and pelargonidine 3-*o*-galactoside, pelargonidine 3-*o*-robinoside, aromadendrin 7-*o*-glucoside, quercetin 3-*o*-galactoside, quercetin 3-*o*-glucuronide, and kaempferol 3-*o*-galactoside [59]	Gallic acid, vanillic acid, caffeic acid, *p*-coumaric acid, trans-ferulic acid, epicatechin, gallate, chlorogenic acid, trans-ferulic cid, and phloridzin [29]	Gallic acid, *p*-hydroxybenzoic acid, catechin, epicatechin, caffeic acid, chlorogenic acid, syringic acid, *p*-coumaric acid, tyrosol, protocatechiuc acid, caftaric acid, coutaric acid, fertaric acid, vanilic acid, syringing acid, procyanidin B2, quercetin-3-*O*-galactoside, kaempferol-3-*O*-rutinoside, rutin isorhamnetib-3-*O*-glucoside, ferulic acid, and quercetin [41,42,57]	Gallic acid, catechin hydrate, chlorogenic acid, caffeic acid, *p*-coumaric acid, trans-ferulic acid, epicatechin gallate, and phloridzin [29]	Catechol, peonidin, catechin 3-*O*-gallate, m-coumaric acid, ferulic acid, mullein, genistein, 4-ethylcatechol, 6-prenylnaringenin, gallic acid, spinacetin, and malvidin 3-*O*-arabinoside [60]
Mineral composition	-	-	-	-	K, Na, Ca, Zn, Mg, Fe, P, Ni, and Mg [61]	K, Na, Ca, Mg, Fe, P, Ni, Mn, and Zn [61,62]	-	-	-	-	-

### 3.2. Oxymel

However, relatively little information exists about the chemical components of oxymels. An oxymel is not always only a mixture of vinegar and honey or sugar: it can be enriched with various medical plants and fruit and vegetable extracts. For example, Abolghasemi et al. [63] found that *Zataria multiflora* Boiss. oxymel contained various chemical compounds. The authors created an oxymel by boiling a mixture of three units of sugar to one unit of vinegar and one unit of water. Gas chromatography/mass spectrometry (GC/MS) demonstrated that the tested oxymel contained terpenes, including carvacrol and thymol, as well as inter alia β-cymene and terpinene. HPLC testing confirmed the presence of caffeic acid (10.9 mg/L), quercetin (652.7 mg/L), *p*-coumaric acid (13.1 mg/L), eugenol (91 mg/L), and rosmarinic acid (116.2 mg/L).

## 4. Biological Activity of Fruit Vinegars

The consumption of fruit vinegars has been found to have pro-health properties in animal and human models.

### 4.1. Antihyperglycemic Effect

Mitrou et al. [64] noted that apple vinegar (30 mL/day) decreased the concentration of blood glucose in humans with type 2 diabetes (n = 11); the authors attributed this to a combination of stimulating glucose uptake and enhancing insulin activity in skeletal muscle. Also, this beneficial action may be due to the presence of acetic acid, which acts via mitogen-activated protein kinase (MAPK). Other studies indicate that other organic acids also counteract the activity of hydrolyzing enzymes such as lactase, maltase, trehalase, and sucrase. Moreover, Ousaaid et al. [61] observed that 2 mL of apple vinegar daily for five weeks decreased the risk of developing hyperglycemia stimulated by a hypercaloric diet in rats.

### 4.2. Antihyperlipidemic Effect

A few papers propose that fruit vinegars may be used to treat dyslipidemia. For example, Bahesheti et al. [65] indicate that supplementation with apple vinegar (30 mL twice a day, for eight weeks) improves lipid profiles, including triglyceride, low-density lipoprotein (LDL), and total cholesterol levels, in patients with hyperlipidemia (n = 19). Also, the consumption of fruit vinegars, including pomegranate vinegar, appears to have similar effects in obese mice and overweight females [17,66,67]. It is possible that this influence on lipid metabolism may take place though the activation of 5′AMP-activated protein kinase (AMPK) by acetic acid in adipose tissue [66].

### 4.3. Antioxidative Effect

Fruit vinegars contain antioxidant phenolic compounds, which may protect against the oxidative stress associated with various diseases. Interestingly, fruit vinegars have been found to demonstrate these properties both in vitro and in vivo. For example, Bouazza et al. [68] reported that vinegars from pomegranate (*Punica granatum* L), prickly pear (*Opuntia ficus-indica* (L.) Mill.), and apple (*Malus domestica* Borkh.) stimulated the activity of various antioxidant enzymes, including glutathione reductase, glutathione peroxidase, and superoxide dismutase, and increased total antioxidant status. However, they also decreased lipid peroxidation, measured by TBARS level, by about 44% in the liver and about 61% in plasma in rats fed a high-fat diet (80 cal/day). In addition, the pomegranate vinegar yielded a significant reduction in lipid profile levels: total cholesterol, 65%; triglycerides, 68%; low-density lipoprotein (LDL), 76%; and atherogenic index, 80%. In this experiment, fifty male Wistar rats were orally dosed with fruit vinegars (7 mL/kg) once daily for 28 weeks. Halima et al. [69] also noted that the consumption of apple vinegar attenuated oxidative stress and reduced the risk of obesity in male Wistar rats receiving a high-fat diet.

### 4.4. Anti-Inflammatory Effect

The results of Wakuda et al. [70] demonstrated that pear vinegar reduced the levels of inflammatory cytokines, including serum interleukin-6 (Il-6) and IL-8, in mice with acute colitis induced by a sodium sulfate (n = 54). Other authors indicate that consumption of apple vinegar inhibits cyclooxygenase-2 (COX-2) in patients with type 2 diabetes [71]. In addition, *Cudrania tricuspidata* fruit vinegar suppressed inflammatory marker expression, decreasing IL-6, TNF-α, MCP-1, iNOS, and nitric oxide (NO), in 3T3-L1 adipocytes and Raw264.7 macrophages in vitro [31,32]. A more detailed review of the anti-inflammatory properties of fruit vinegars is given in a review by Ousaaid et al. [72], which discusses their use in the prevention and treatment of various inflammatory diseases including colitis, arthritis, atopic dermatitis, and asthma; the authors note that artisanal fruit vinegars are transformed into anti-inflammatory compounds in the digestive system more effectively than commercial varieties, and their properties may promote anti-inflammatory activity by influencing pro-inflammatory cytokine production and altering the intestinal microbiota. In addition, Abdali et al. [37] reported that supplementation of apple vinegar (10 mL/kg) reduced carrageenan-stimulated inflammation by about 37% in Wistar rats. The dosing also appeared to exert antidepressant properties, reducing immobility time by about 30% among the rats.

### 4.5. Other Biological Properties

Recently, the results of Seyidoglu et al. [73] demonstrated that all used hawthorn vinegars, especially ultrasound-treated vinegar, had positive actions on intestinal health and boosted immunity in Wistar albino rats (n = 56). In this study, the authors used traditional production of vinegar (0.5 mL/kg bw), thermal pasteurization of vinegar (0.5 and 1 mL/kg bw), and ultrasound treatment of vinegar (0.5 and 1 mL/kg bw). They were administered by oral gavage daily (for 45 days). Various other authors [74,75] suggest that a possible mechanism of hawthorn vinegar could be related to phenolic compounds.

Na et al. [76] noted that fruit vinegar decreased blood pressure in spontaneously hypertensive rats, which they attributed to the downregulation of angiotensin II receptor type 1 (AT1R) expression via the AMPK/PGC-1α-PPARγ pathway. Similar changes in protein expression were also found in SVAREC cells treated with 200 or 400 μmol/L acetate; the authors therefore suggested that the antihypertensive effects of the used vinegar may be due to its acetic acid content. Recently, Tang et al. [23] found that Fardh vinegar, made from the fruit of the date palm (*Pheonix dactylifera* L.), inhibited angiotensin-converting enzyme 2 (ACE2) activity in vitro and displayed good antioxidant activity by radical scavenging. Ali et al. [18,19] and Li et al. [77] also found that vinegars from red and black date fruits demonstrated antioxidant potential in vitro.

Tripathi and Mazumder [78] found that apple vinegar inhibited the activity of monoamine oxidase (MAO). This may have a protective effect against Alzheimer’s disease as MAO catalyzes the oxidative deamination of amines in the brain and peripheral tissues, whose buildup plays a part in AD. It is also involved in the metabolism of monoamines and is vital for cognition.

Apple vinegar and other fruit vinegars have been found to have in vitro antimicrobial properties against bacteria, such as *Bacillus subtilis*, *Escherichia coli*, *Salmonella*, and *Staphylococcus aureus* [37,79,80,81], and against certain fungi, including *Candida albicans* spp. [37,79,82]. Some papers indicate that these vinegars may also improve skin barrier integrity in atopic dermatitis [83,84,85].

A recent study by Yim et al. [86] found that mulberry vinegar intake (1 mg/kg bw) preserved bone mineral density in ovariectomized rats, mainly by inhibiting osteoclastic activity. They suggested that this vinegar may develop as a functional food for anti-osteoporosis in menopausal females. An in vitro study by Bang et al. [87] found that mulberry vinegar (10–100 µg/mL) prevented neuroinflammation in C6 glial cells stimulated by lipopolysaccharide (LPS)/interferon-γ (IFN-γ) by regulating the NF-κB signaling pathway; treatment inhibited COX-2 and iNOS production and increased glial activation, through the downregulation of ionized calcium-binding adapter molecule-1 (Iba-1) and glial fibrillary acidic protein (GFAP).

It has also been found that apple vinegar supplementation (1 mL/kg/day, for five weeks) ameliorated changes in blood platelets, blood cell count, mean corpuscular volume, hemoglobin concentration, and mean capsulated hemoglobin induced by phenylhydrazine in Wistar rats [2]. Shams et al. [88] examined the effect of apple vinegar on fallicuogenesis and ovarian kisspeptin in rats with non-alcoholic fatty liver disease caused by a high-fat diet (n = 28). Supplementation of apple vinegar was found to increase ovarian kisspeptin expression and raise primordial, estradiol, and small primary follicles. In addition, the used vinegars demonstrated anti-lipidemic and anti-glycemic properties.

Recent data indicate that orange vinegar and its main phenolic compounds (coumaric acid, epicatechin, and catechin) reduce the levels of pro-inflammatory factors, reactive oxygen species, and NADPH in Caco-2 cells. They have also been found to protect against the accumulation of advanced glycation end products, which play roles in the development of various degenerative disorders [24].

The biological activities of various fruit vinegars, including their anti-inflammatory, antioxidant, and neuroprotective properties, identified in animal and human models are presented in more detail in Table 2. These vinegars may also counteract the development of various disorders, including diabetes and cardiovascular diseases. Recent studies also suggest that apple vinegar may offer benefits in treating insulin resistance, osteoporosis, and certain neurological diseases such as Alzheimer’s disease; it may also support weight loss. The biological properties of fruit vinegars and their potential molecular mechanisms are also presented in Figure 2. In addition, this figure demonstrates modes of action of vinegar components, including phenolic compounds, vitamins, and acetic acid. For example, phenolic compounds and vitamins exhibit antioxidant properties. Acetic acid was found to suppress body fat accumulation, and it may determine the anti-obesity and cardioprotection properties of fruit vinegars.

## 5. Biological Activity of Oxymels

Oxymels also appear to possess pro-health properties and may also serve as effective and safe treatment options for managing various diseases, including obesity, metabolic syndrome, and asthma. Numerous preclinical and clinical studies indicate the beneficial action of oxymels [90,92,104]. For example, Abolmaali et al. [92] observed that oxymel containing vinegar, honey, and squill bulb, at doses of 100, 200, and 400 mg/kg, decreased the severity of seizure induced by penta-lenetetrazole in mice (n = 39) compared with controls; it also reduced the mortality rate of the animals in a dose-dependent manner.

It was also found that 0.1 mL of warm oxymel consisting of vinegar and sugar, once per day for 10 days, was also found to decrease IL-4 gene expression, perivascular and peribronchial inflammation, and hypersecretion in mice with asthma (n = 24) [90]. Squill oxymel containing honey, vinegar, and *Drima martima* (L.) stem, administered at 10 mL, twice a day, appeared to be a safe and effective treatment in patients with moderate to severe persistent asthma (n = 60) [102], as was squill oxymel, 10 mL twice a day for four weeks, in patients with chronic obstructive pulmonary disease (n = 42) [103].

Other results indicate that oxymel composed of vinegar, sugar, and thyme (300 or 500 mg/kg body weight (BW)/day, for 12 weeks) ameliorates obesity in Sprague–Dawley rats (n = 80) [93]. The authors suggest that it exerts its anti-obesity effects by improving lipid metabolism, inflammation, and oxidative stress, as indicated by serum and hepatic TBARS level and antioxidant enzyme activity. It also appears that the oxymel may also regulate the expression of sterol regulatory element-binding transcription factor (SREBP), carnitine palmitoyl transferase I (CPT-1), nuclear factor kappa B (NF-κB), and erythroid 2-related factor 2 (Nrf-2) on the genetic level.

Sarbaz Hoseini et al. [91] found oxymel (1 mL/day, for eight weeks) to have anti-diabetic effects in Wistar rats with type 2 diabetes and *Zataria* oxymel (0.75 mg *Zataria multiflora* Boiss. in 10 mL oxymel) to reduce insulin resistance in overweight patients (n = 200). Another study found oxymel (10 mL/kg BW, orally, for 14 days) to change the lipid profile in Wistar rats with hyperlipidemia (n = 30) [89].

*Berberis vulgaris* oxymel also appears effective in the treatment of patients with refractory primary sclerosing cholangitis and primary biliary cholangitis (n = 87). The tested oxymel did not have any apparent adverse effect on the kidney [99]. However, oxymel did not appear to demonstrate any significant effects on blood pressure in healthy volunteers [95,96].

More details about different biological activities of oxymels in animal and human models are given in Table 2. Interestingly, in most cases, oxymels have been studied on models of various diseases, including asthma, obesity, and migraine; comparatively few have been conducted on healthy people (Table 2). The most frequently tested type was simple oxymel, i.e., consisting of vinegar and a sweet component (honey or sugar), with squill oxymel being less common. Some other types also contain *Z. multiflora*, *C. spinose*, and *B. vulgaris*. Most importantly, like the fruit vinegars, none of the used oxymels had any adverse effects on animals or humans.

Recently, oxymel has been used successfully on wounds as a topical application against antibiotic-resistant bacteria. While both vinegar and honey have been used historically as antiseptics, the combination was found to be 1000 times more effective at killing bacteria than vinegar alone and 100,000 times more than honey alone [27,63].

Although the mechanism behind the antimicrobial action of oxymel remains unclear, it is likely that it acts on a number of different levels simultaneously. It is also possible that the different ingredients, e.g., vinegar and its main component, acetic acid, phenolic compounds, and sweet components, may demonstrate synergy when applied together. Furthermore, the biological effects of the oxymel may be further enhanced by the presence of plant components.

## 6. Conclusions

For the first time, this review paper demonstrates that not only fruit vinegars but also oxymels appear to be good candidates for dietary supplements with beneficial effects on chronic conditions, such as obesity and asthma. They are also easy to administer: oxymels can be taken by the spoonful or added to beverages, such as water or tea, and the combination of honey and vinegar has a pleasant taste. Fruit vinegars and oxymels are also less expensive than other supplements or functional foods and can even be prepared at home.

The multidirectional effects of fruit vinegars and oxymels result from the synergy of different chemical compounds, including organic acids (mainly acetic acid), phenolic compounds, vitamins, minerals, and fermentation products (esters, aldehydes, and ketones). Together, these give the preparation its characteristic taste and aroma [25,26,46,105]. However, more studies are needed to understand the interactions between all the different components, not only the phenolic compounds and organic acids. In addition, more research is needed on their mechanisms of action. Although no serious side effects have been noted to date, further studies with large sample sizes are needed to understand the possible side effects of long-term fruit vinegar and oxymel use. These studies should include healthy people and those with various diseases, in addition to animals, and should examine their potential interactions with functional foods, supplements, and drugs.

## Figures and Tables

**Figure 1 ijms-26-00007-f001:**
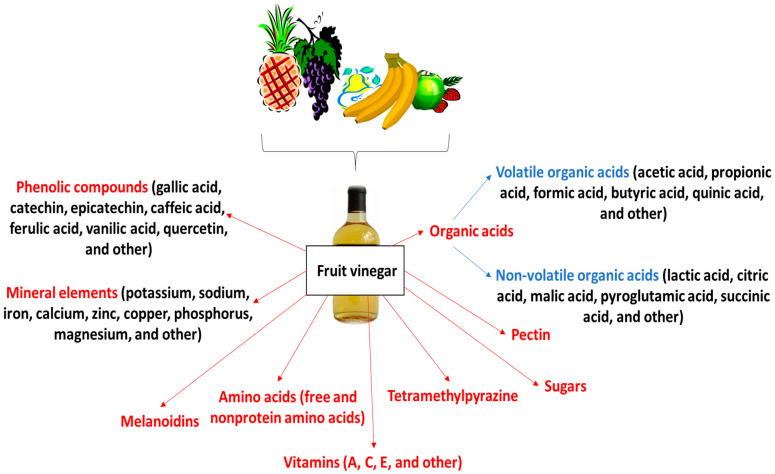
Chemical characteristic of fruit vinegars.

**Figure 2 ijms-26-00007-f002:**
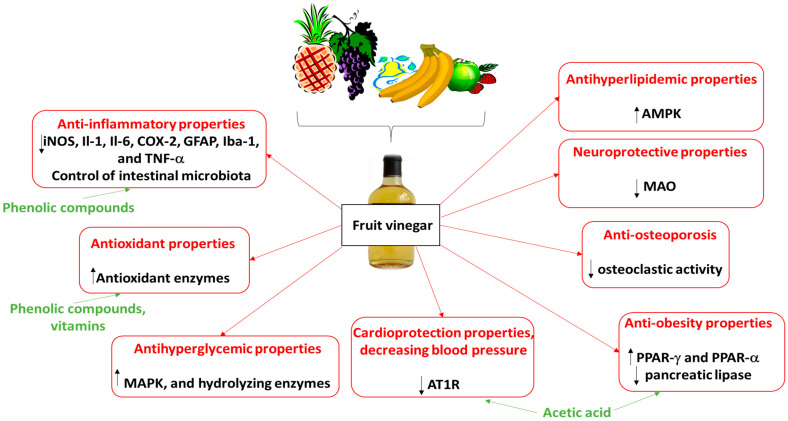
**Biological properties of fruit vinegars and their potential molecular mechanisms.** AMPK, 5′AMP-activated protein kinase; AT1R, angiotensin II receptor type 1; COX, cyclooxygenase; GFAP, glial fibrillary acidic protein; Il, interleukin; Iba-1, ionized calcium-binding adapter molecule-1; iNOS, inducible nitric oxide synthase; MAO, monoamine oxidase; MAPK, mitogen-activated protein kinase; PPAR, peroxisome proliferators-activated receptors; TNF-α, tumor necrosis factor-α.

**Table 2 ijms-26-00007-t002:** The various biological activities of fruit vinegars and oxymels, based on animal and human models.

Fruit Vinegar/Oxymel	Dosage	Experimental Model	Biological Activity and Side Effects	References
	Animal Model			
Apple vinegar	2 mL daily of apple vinegar, for five weeks	Fed rats with hypercaloric diet	Antihyperglycemic effect; -	[61]
Apple vinegar	0.7% daily, for three months	Murine model of Alzheimer disease	Neuroprotective effect; -	[78]
Apple vinegar	1 mL/kg/day, for five weeks	Wistar rats treated with phenylhydrazine	Anti-anemic effect (ameliorating changes in blood platelets and hemoglobin concentration); -	[28]
Apple vinegar	5 g vinegar powder/100 g standard diet, for 8 weeks	Rats treated for high-fat diet-induced nonalcoholic fatty liver diseases	Anti-glycemic and anti-lipidemic effect; -	[88]
Apple vinegar	10 mL/kg	Wistar rats	Anti-inflammatory, antidepressant, anti-bacterial, and anti-fungal effects; -	[37]
Mulberry vinegar	1 mg/kg bw, for 12 weeks	Ovariectomized rats	Anti-osteoporosis effect (inhibiting osteclastic activity); -	[86]
Pear vinegar	4.5% daily, for six days	Dextract sodium sulfate-induced acute colitis mouse model	Anti-inflammatory effect (reducing the levels of inflammatory cytokines); -	[70]
Pomegranate, prickly pear, and apple vinegars	7 mL/kg of fruit vinegars once daily for 28 weeks	High-fat-fed rats (80 cal/day)	Antioxidant effect (stimulating the activity of various antioxidant enzymes, increasing total antioxidnat status, inhibiting lipid peroxidation); -	[68]
Hawthorn vinegar	Traditional production of vinegar (0.5 mL/kg bw), thermal pasteurization of vinegar (0.5 and 1 mL/kg bw), and ultrasound treatment of vinegar (0.5 and 1 mL/kg bw) for 45 days	Wistar albino rats	Positive action on intestinal health and boosting immunity; -	[73]
**Human model**				
Apple vinegar	30 mL/day	Humans with type 2 diabetes	Antihyperglycemic effect (stimulatimg glukose uptake, enhancing inslulin activity in skeleton muscle); -	[64]
Apple vinegar	30 mL twice a day	Patients with hyperlipidemia	Antihyperlipidemic effect (improving lipid profile: triglyceride, LDL, and total cholesterol); -	[65]
**Animal model**				
Oxymel	10 mL/kg BW, orally, for 14 days	Wistar rats with hyperlipidemia	Changing lipid profile; no side effect	[89]
Oxymel	0.1 mL of warm oxymel/day, for 10 days	Mice with asthma	Decreasing IL-4 gene expression, perivascular and peribronchial inflammation, and hypersecretion; no side effect	[90]
Oxymel	1 mL/day, for 8 weeks	Wistar rats with type 2 diabetes	Anti-diabetic effect (reducing insulin resistance); no side effect	[91]
Squill oxymel	100, 200, and 400 mg/kg	Mice (penty-lenetetrazole-induced seizure)	Decreasing the denation of seizure in mice (penty-lenetetrazole-induced seizure), and oxymel reduced the mortality rate of animals in a dose-dependent manner; no side effect	[92]
Thyme oxymel	300 or 500 mg/kg BW/day, for 12 weeks	Sprague–Dawley rats with obesity	Decreasing obesity (improving lipid metabolism, inflammation, and oxidative stress); no side effect	[93]
**Human model**				
Oxymel	2 tablespoons with 250 mL water, daily, for four weeks	Healthy volunteers	No significant effect on blood pressure; -	[94,95]
Oxymel	400 mL/day, for 30 days	Healthy volunteers	No significant effect on blood pressure; no side effect	[96]
Oxymel	2 tablespoons with 250 mL water, daily, for four weeks	Healthy volunteers	No significant effect on lipid profile; -	[94,95]
Oxymel	30 mL/day, for 30 days	Overweight and obese patients	A significant positive action on serum cholesterol and body weight; no side effect	[97]
Oxymel	0.75 g or 1.5 g of Z. multiflora Boiss. in 10 mL oxymel, 10 mL twice daily, for 12 weeks	Overweight patients	Improved metabolic parameters, for example, insulin resistance; -	[92]
Oxymel	200 mL/day, for 60 days	Patients with migraine with or without aura	No significant effect on frequency and duration of headaches; -	[98]
*B. vulgaris* oxymel	0.5 mL/kg/day, twice daily, for three months	Patients with refractory primary sclerosing cholangitis and primary biliary cholangitis	A significant attenuation of aspartate transaminase, alanine transaminase, alkaline phosphatase, gamma-glutamyl transferase, and direct and total bilirubin; no side effect	[99]
*C. spinosa* oxymel	10 mL/day, for three months	Patients with type 2 diabetic and metabolic syndrome	A significant decrease in weight and BMI; no side effect	[100]
Squill oxymel	10 mL/day, for eight weeks	Patients with knee osteoarthritis	A significant positive action on treatment of patients; no side effect	[101]
Squill oxymel	10 mL twice daily, for six weeks	Patients with moderate to severe asthma	A significant positive action on treatment of patients; -	[102]
Squill oxymel	10 mL twice daily, for four weeks	Patients with chronic obstructive pulmonary disease	A significant positive action on O_2_ saturation; no side effect	[103]

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
