# Peer review of "Pro-Health Potential of Fruit Vinegars and Oxymels in Various Experimental Models"

_ijms, 2024, doi:10.3390/ijms26010007_

Round 1

Reviewer 1 Report

Comments and Suggestions for Authors

This review article by Olas aims to discuss the current state of knowledge on chemical composition and biological activity with mechanisms of vinegars and oxymels. This topic is of interest and should provide several future perspectives in this field. However, the following points need to be addressed for a possible publication in IJMS.

1.    The abstract should be rewritten to be more realistic based on the elucidations of topics discussed in this review article, instead of only highlighting the topics inside the review article.

2.    Another keyword “submerged fermentation” should be added to keywords.

3.    Throughout the manuscript there are many 1 or 2 sentence paragraphs which must to be consolidated into several large paragraphs.

4.    “no data” in tables should be replaced with a dash symbol and referring in the footnote as ‘data not available’.

5.    In section 2 the author has mentioned that this review article includes published reports from the last 20 years, but most of the articles discussed in sections 3-5 are within 10-12 years, with most of them are recent ones. The authors should clarify and mention the correct year range considered for this review.

6.    The whole manuscript including references and their citations in the text needs to conform to the journal guidelines.

7.    The sections 3-5 should be elaborated with in-depth discussions providing several sub-topics under each of section 3, 4 and 5.

8.    Both Table 2 should be provided with some details of the experiments, results and conclusion for each citation.

9.    Some important figures and tables (at least 2 each) from the published articles should be reproduced with permission from the publishers. Mostly they are free to obtain.

Author Response

This review article by Olas aims to discuss the current state of knowledge on chemical composition and biological activity with mechanisms of vinegars and oxymels. This topic is of interest and should provide several future perspectives in this field. However, the following points need to be addressed for a possible publication in IJMS.

Thank you for reviewing the manuscript and providing such helpful comments. All of them have been taken into consideration when revising the manuscript.

  1. The abstract should be rewritten to be more realistic based on the elucidations of topics discussed in this review article, instead of only highlighting the topics inside the review article.

Response: I have changed abstract. Now, it is: „Fruits are excellent sources of substrate for various fermented products, including fruit vinegars, which are typically produced by submerged fermentation. Some evidence suggests that fruit vinegar consumption can alleviate certain disorders, including hyperlipidemia, inflammation and hyper-glycemia. Fruit vinegars have also bacteriostatic and antihypertensive actions. Recent studies also suggest that apple vinegar may offer benefits in treating insulin resistance, osteoporosis and certain neurological diseases such as Alzheimer’s disease; it may also support weight loss. Recent studies in animal and human models have considerably broadened our understanding of the biological properties of not only fruit vinegars, but also oxymels, i.e. mixtures of vinegar and honey or sugar. This paper reviews the current state of knowledge regarding vinegars and oxymels, with a special emphasis on their chemical composition and the mechanisms behind their biological activity and pro-health potential. The multidirectional effects of fruit vinegars and oxymels result from the synergy of different chemical compounds, including organic acids (mainly acetic acid), phenolic compounds, vitamins, minerals, and fermentation products. However, more studies are needed to understand the interactions between all the different components, not only the phenolic compounds and organic acids. In addition, more research is needed on their mechanisms of action. Although no serious side effects have been noted to date, further studies with large sample sizes are needed to understand the possible side effects of long-term fruit vinegar and oxymel use.”

  1. Another keyword “submerged fermentation” should be added to keywords.

Response: I have added new keyword “submerged fermentation”.

  1. Throughout the manuscript there are many 1 or 2 sentence paragraphs which must to be consolidated into several large paragraphs.

Response: I have corrected it.

  1. “no data” in tables should be replaced with a dash symbol and referring in the footnote as ‘data not available’.

Response: I have corrected it.

  1. In section 2 the author has mentioned that this review article includes published reports from the last 20 years, but most of the articles discussed in sections 3-5 are within 10-12 years, with most of them are recent ones. The authors should clarify and mention the correct year range considered for this review.

Response: I have corrected it. Now, it is: “A literature search was performed of PubMed, Science Direct, Scopus, Springer, Web of Knowledge, Web of Science, and Google Scholar, using various combinations of the keywords: “fruit vinegar”, “oxymel”, “biological activity”, and “pro-health potential”. No time criteria were applied to the search, but recent papers were evaluated first. The review included book chapters, review papers and research papers. The abstracts of any identified articles were initially analyzed to confirm whether they met the inclusion criteria. Any relevant identified articles were summarized. After obtaining the full texts of the included studies, the reference sections were also manually examined to identify any additional new articles.”.

  1. The whole manuscript including references and their citations in the text needs to conform to the journal guidelines.

Response: I have corrected it.

  1. The sections 3-5 should be elaborated with in-depth discussions providing several sub-topics under each of section 3, 4 and 5.

Response: I have added sub-topics under each section 3 and 4. I have not decided to add sub-topics under section 5, because there is only few papers about biological properties of oxymels. However, I have described more information about biological properties of not only vinegars, but also oxymels in table 2.

  1. Both Table 2 should be provided with some details of the experiments, results and conclusion for each citation.

Response: I have added more information about it in table 2. In addition, I have also added more information in the text of manuscript.

  1. Some important figures and tables (at least 2 each) from the published articles should be reproduced with permission from the publishers. Mostly they are free to obtain.

Response: I prepared the tables and figures myself, based on the results described by different authors.

Reviewer 2 Report

Comments and Suggestions for Authors

The study presented in the paper “Pro-health potential of fruit vinegars and oxymels in various experimental modelsreviews the current state of knowledge regarding vinegars and oxymels, with a special emphasis on their chemical composition and the mechanisms behind their biological activity and pro-health potential.

The review is interesting, the methodology is adequate and explicitly stated and the subject is very topical. The manuscript is clear, relevant for the field and presented in a well-structured manner.

Even if there are various reviews about fruit vinegars, this review is relevant and of interest to the scientific community.

The results and conclusions are remarkable and for this reason, I recommend the publication of this study after a minor revision.

The authors are invited to clarify the following aspects:

Ø  The novelty aspect needs to be clearly articulated. What parts do you consider original or relevant for the field? What specific gap in the field does the review address?

Ø  Please check the font of last paragraph of page 2 and table 2.

Ø  Figure 2 is a little bit difficult to interpret and understand, and is not clear explained in the text.

Ø  I would suggest that in the conclusions include some final considerations on the novelties that this work provides with respect to others already existing in the bibliography.

Overall, this work is of scientific interest and is relevant within its scientific area.

Author Response

The study presented in the paper “Pro-health potential of fruit vinegars and oxymels in various experimental models” reviews the current state of knowledge regarding vinegars and oxymels, with a special emphasis on their chemical composition and the mechanisms behind their biological activity and pro-health potential.

The review is interesting, the methodology is adequate and explicitly stated and the subject is very topical. The manuscript is clear, relevant for the field and presented in a well-structured manner.

Even if there are various reviews about fruit vinegars, this review is relevant and of interest to the scientific community.

The results and conclusions are remarkable and for this reason, I recommend the publication of this study after a minor revision.

Thank you for reviewing the manuscript and providing such helpful comments. All of them have been taken into consideration when revising the manuscript.

The authors are invited to clarify the following aspects:

Ø  The novelty aspect needs to be clearly articulated. What parts do you consider original or relevant for the field? What specific gap in the field does the review address?

Response: I have changed the aim of this review, and conclussion. Now, it is: „Recently, few papers have described chemical characteristics and biological properties of various vinegars [Xia et al., 2020; Pusaiid et al., 2022; Chen et al., 2023; Khalifa et al., 2024]. However, Ousaaid et al. [2022] only demonstrated the chemical composition of fruit vinegars and their therapeutic application. In addition, only one systematic review on oxymel has been published [Darani et al., 2023]. The present work reviews the up-to-date literature concerning the pro-health potential of fruit vinegars, especially filtered and pasteurized clear vinegars, and oxymels; it also examines their chemical composition and the mechanisms behind their biological properties. Of course, artisanal vinegars, i.e. live vinegars, are also on sale, but no comprehensive literary data on their health-promoting potential currently exists.”

„The first time, this review paper demonstrates that not only fruit vinegars, but also oxymels ap-pear good candidates for dietary supplements with beneficial effects on chronic conditions, such as obesity and asthma. They are also easy to administer, oxymels can be taken by the spoonful or added to beverages, such as water or tea, and the combination of honey and vinegar has a pleasant taste. Fruit vinegars and oxymels are also less ex-pensive than other supplements or functional foods and can even be prepared at home.”

Ø  Please check the font of last paragraph of page 2 and table 2.

Response: I have corrected it.

Ø  Figure 2 is a little bit difficult to interpret and understand, and is not clear explained in the text.

Response: I have not changed Figure 2. However, I have added more information about in the text of manuscript. Now, it is: „The biological properties of fruit vinegars and their potential molecular mechanisms are also presented in Figure 2. In addition, this figure demonstrates modes of action of vinegar components, including phenolic compounds, vitamins, and acetic acid. For example, phenolic compounds and vitamins exhibit antioxidant properties. Acetic acid was found to suppress body fat accumulation, and it may decide about anti-obesity and cardioprotection properties of fruit vinegars.”.

Ø  I would suggest that in the conclusions include some final considerations on the novelties that this work provides with respect to others already existing in the bibliography.

Response: I have changed the aim of this review, and conclussion. Now, it is: „Recently, few papers have described chemical characteristics and biological properties of various vinegars [Xia et al., 2020; Pusaiid et al., 2022; Chen et al., 2023; Khalifa et al., 2024]. However, Ousaaid et al. [2022] only demonstrated the chemical composi-tion of fruit vinegars and their therapeutic application. In addition, only one systematic review on oxymel has been published [Darani et al., 2023]. The present work reviews the up-to-date literature concerning the pro-health potential of fruit vinegars, especially filtered and pasteurized clear vinegars, and oxymels; it also examines their chemical composition and the mechanisms behind their biological properties. Of course, artisanal vinegars, i.e. live vinegars, are also on sale, but no comprehensive literary data on their health-promoting potential currently exists.”

„The first time, this review paper demonstrates that not only fruit vinegars, but also oxymels ap-pear good candidates for dietary supplements with beneficial effects on chronic condi-tions, such as obesity and asthma. They are also easy to administer, oxymels can be taken by the spoonful or added to beverages, such as water or tea, and the combination of honey and vinegar has a pleasant taste. Fruit vinegars and oxymels are also less ex-pensive than other supplements or functional foods and can even be prepared at home.”

Overall, this work is of scientific interest and is relevant within its scientific area.

Round 2

Reviewer 1 Report

Comments and Suggestions for Authors

The authors have satisfactorily addressed all the comments raised by reviewers and substantially improved the overall quality of the article. Therefore, I recommend accepting this article for publication in IJMS.